# Understanding Self-Attention of Self-Supervised Audio Transformers

**Shu-wen Yang** [1]   **Andy T. Liu** [1 2]   **Hung-yi Lee** [1 2]

## Abstract

Self-supervised Audio Transformers (SAT) enable great success in many downstream speech applications like ASR, but how they work has not been widely explored yet. In this work, we present multiple strategies for the analysis of attention mechanisms in SAT. We categorize attentions into explainable categories, where we discover each category possesses its own unique functionality. We provide a visualization tool for understanding multi-head self-attention, importance ranking strategies for identifying critical attention, and attention refinement techniques to improve model performance.

## 1. Introduction

Adapting the idea of self-supervised learning (Devlin et al., 2018; Yang et al., 2019; Liu et al., 2019; Lan et al., 2019) to continuous speech has received much attention in recent work (Liu et al., 2020; Jiang et al., 2019; Song et al., 2019; Wang et al., 2020; Baevski et al., 2019b;a), where Transformer Encoders with multi-head self-attention (Vaswani et al., 2017) are pre-trained on a large amount of audio data in a self-supervised scheme. Once pre-trained, they are used to improve various downstream supervised tasks, including phone classification, speaker recognition, SLU, and ASR. Despite the great success of these Self-supervised Audio Transformers (SAT)[1], their internal attention are often neglected and not explored, as we have little understanding of how they work, or the knowledge they acquire from a large amount of unlabeled data. Understanding how SAT models draw conclusions is crucial for both their improvement

and application. In the area of natural language processing (NLP), explaining and interpreting pre-trained black-box models like BERT have been a well-explored topic (Aken et al., 2020; Hao et al., 2019; Kovaleva et al., 2019; Clark et al., 2019; Tenney et al., 2019a;b). However, the analysis of models that are pre-trained on speech has not seen such widespread exploration, and remains an important and challenging endeavor for the speech community.

In this work, we propose to analyze the multi-head self-attention mechanism of SAT through the following methods: visualization, categorization, functionality study, and importance ranking. We found that the self-attentions of SAT models tend to converge into three categories: global attentions, vertical attentions, and diagonal attentions. Diagonal attentions either highly attend to $\pm t$ neighbor or are highly correlated with phoneme boundaries; vertical attentions often concentrate on specific phonemes. As for noisy global attentions, we provide a visualization tool to draw insights about their implicit operations. Through our quantized ranking analysis, we conclude that diagonal attentions outrank the most in terms of importance, followed by vertical attentions. Last but not least, we introduce attention refinement methods which allow us to improve learned representations by partially removing global attentions or constraining attention span, resulting in a faster inference time and higher performance.

## 2. Self-Supervised Audio Transformers

The main ideology of NLP BERT pre-training (Devlin et al., 2018; Yang et al., 2019; Liu et al., 2019; Lan et al., 2019) is to corrupt the input word tokens by randomly masking or permuting them with a probability policy, layers of Transformer Encoder (Vaswani et al., 2017) are trained together with a classifier that estimates the masked words at the output. Primarily inspired by this idea, previous works (Liu et al., 2020; Song et al., 2019; Jiang et al., 2019; Wang et al., 2020; Baevski et al., 2019b) proposed self-supervised learning for audio with Transformer Encoders. In this work, we refer to these types of models as Self-Supervised Audio Transformers, SAT. Unlike BERT where the inputs are discrete text tokens, the inputs of SATs are acoustic features (e.g., MFCC, FBANK, Mel-Spectrogram), which form much longer sequences and could be extremely similar to

---

[1]College of Electrical Engineering and Computer Science, National Taiwan University [2]Graduate Institute of Communication Engineering, National Taiwan University. Correspondence to: Shu-wen Yang <r08944041@ntu.edu.tw>, Andy T. Liu <f07942089@ntu.edu.tw>, Hung-yi Lee <hungyilee@ntu.edu.tw>.

*Published at the workshop on Self-supervision in Audio and Speech at the $37^{th}$ International Conference on Machine Learning*, Vienna, Austria. Copyright 2020 by the author(s).

[1]These pre-trained transformer encoders have several different names in their original papers. In this paper we refer to them as SAT for simplicity.

their neighbor features since speech signal is continuously varying. Some SATs take continuous acoustic features as input directly (Liu et al., 2020; Song et al., 2019), while some conduct vector quantization in advance (Baevski et al., 2019b;a). Also, different from BERT where the model is trained by estimating discrete tokens, SATs change to minimize reconstruction error between the real frame and the predicted frame (Liu et al., 2020; Jiang et al., 2019) or classification error for the real frame among sampled distracting frames (Baevski et al., 2019b;a).

Among all the variants of SATs, we address our focus on SATs that take continuous acoustic features as input with reconstruction loss. In our analysis, we particularly follow the framework described in Mockingjay (Liu et al., 2020). In Mockinjay, two techniques of downsampling and consecutive masking are introduced to resolve these issues. Downsampling is applied on input features to adapt SATs to long sequences. To reduce the length of frames by a factor of $R_{factor}$, consecutive frames of $R_{factor}$ amount are reshaped and stacked into one frame (Sperber et al., 2018; Pham et al., 2019). On the other hand, consecutive masking is applied during pre-training to avoid the model from exploiting the local smoothness of acoustic frames. Instead of masking a single frame, consecutive frames of $C_{num}$ are masked to zero. To study the attentions of SAT models, we use the prevailing framework of the LARGE model described in Mockingjay, which consists of 12 layers of Transformer Encoders. We train three models on the LibriSpeech (Panayotov et al., 2015) train-clean-360 subset with identical settings as in Mockingjay, except for $C_{num} \in \{3, 6, 9\}$, and we name them as M3, M6, M9.

## 3. Notations

We first define notation for self-attention mechanism and SAT representations. Given a length $T$ sequence of vectors $\boldsymbol{x} = x_1, ..., x_T \in \mathbb{R}^d$, we denote $A_u^h \in \mathbb{R}^{T \times T}$ as attention weights for all query-key pairs of a head $h$ when propagating an utterance $u$. Hence, $A_u^h[q, k] \in \mathbb{R}$ is the attention weight of $x_q$ attending to $x_k$. We use $q$ for timestamp of query; $k$ for timestamp of key, where $1 \leq q, k \leq T$. As a result, $A_u^h[q] \in \mathbb{R}^T$ is the attention distribution formed by $x_q$, which is a row if we view $A_u^h$ as a map. When analyzing the representations of a $L$-layer SAT, we denote $\boldsymbol{x^l} = x_1^l, ..., x_T^l \in \mathbb{R}^d$ as the representations of a given layer, where $0 \leq l \leq L$ and $\boldsymbol{x^0}$ represents input features.

## 4. Visualization and Categorization

We plot out $A_u^h \in \mathbb{R}^{T \times T}$ as an attention map, where $A_u^h[0, 0]$ starts from the upper-left corner, like Fig 1[2]. SAT

[2]Supplementary materials: https://github.com/leo19941227/Self-Attention-on-SATs

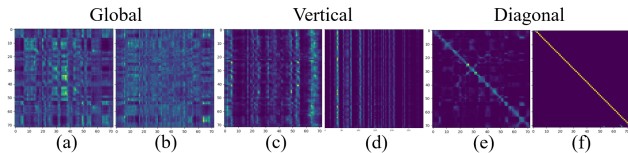

Global       Vertical       Diagonal

(a)    (b)    (c)    (d)    (e)    (f)

*Figure 1.* Attention maps of heads favored by G, V, D, visualized with the same utterance. (a)(c)(e) are average cases; (b)(d)(f) are extreme cases found by maximizing the metrics.

attentions tend to converge into three categories: (1) *global*: flat attention distributions; (2) *vertical*: attention maps with vertical lines, and (3) *diagonal*: attention maps with clear diagonal. Because attention maps of a head are similar across utterances with respect to the three categories, we study self-attention on the basis of head instead of a single attention map. To classify heads into three categories, we define three metrics to quantify a head $h$: globalness $G$, verticality $V$ and diagnality $D$ in equations 1, 2, 3, respectively.

$$G(h) = \mathbb{E}_{u \sim U} \left[ \frac{1}{T} \sum_{q=1}^{T} \mathbb{H}( A_u^h[q] ) \right] \qquad (1)$$

$$V(h) = \mathbb{E}_{u \sim U} \left[ -\mathbb{H}( \frac{1}{T} \sum_{q=1}^{T} A_u^h[q] ) \right] \qquad (2)$$

$$D(h) = \mathbb{E}_{u \sim U} \left[ -\frac{1}{T^2} \sum_{q=1}^{T} \sum_{k=1}^{T} |q - k| \cdot A_u^h[q, k] \right] \quad (3)$$

where $\mathbb{H}$ is the standard definition of entropy, and $U$ is a speech corpus. Based on G, V, D, we would have three ranking lists for all heads. If among the three ranking lists, a head has the highest rank based on the list of G, it would be categorized as global, and so on. We use ranking instead of values because the metrics may not have the same numerical scale. Fig 1 shows two attention maps for each category.

Diagonal attentions attend to local neighbors for every query. Some exhibit a highly focused behavior like Fig 1(f) and some are block diagonal like Fig 1(e). Interestingly, no SAT contains highly focused diagonal attention at main diagonal. They shift either to the left or right, and larger masking span $C_{num}$ is accompanied by a larger shift, possibly due to SAT models trying to get useful information from further frames. The functionality of block diagonal attentions is discussed in section 5. Vertical attentions like Fig 1(c)(d) always attend to similar locations for all queries given an utterance; global attentions like Fig 1(a)(b) behave randomly. These two categories are discussed in section 6. Finally, we visualize the head distribution[2] according to metrics and find the model trained with a larger masking span $C_{num}$ has more global heads. On the contrary, M3 contains the most diagonal heads, suggesting that smaller $C_{num}$ makes SAT focus on local structure more.

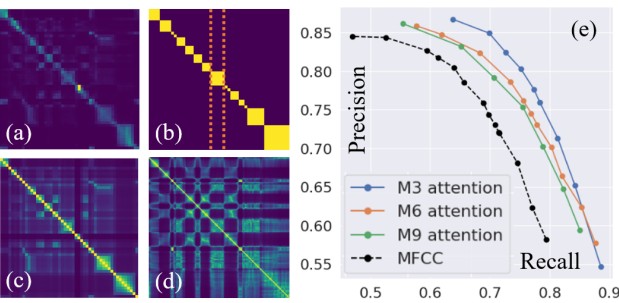

*Figure 2.* Four images on the left side are plotted with the same utterance. (a) a block diagonal attention map. (b) a block diagonal map plotted with true phoneme boundaries. Two orange dotted lines show two examples of boundaries. (c) similarity matrix for (a). (d) similarity matrix for MFCCs. (e) precision-recall curve for M3, M6, M9 attentions and MFCCs.

## 5. Phoneme Segmentation

There are attention maps of clear block diagonal like Fig 2(a). The borders of blocks might be phoneme boundaries, as illustrated in Fig 2(b). It seems diagonal attention knows phoneme intervals. We conduct phoneme segmentation to examine the correlation.

We mainly follow the algorithm proposed in (Bhati et al., 2017), which first calculates a similarity matrix from a sequence of features, containing all pairwise distances between features, and then extract boundary points from the similarity matrix. For segmentation with an attention map, its rows are considered as the feature sequence for computing the similarity matrix[2]. Examples of similarity matrices are shown in Fig 2(c)(d) when segmentation features are the attentions map and MFCCs, respectively. We slightly modify the boundary-point-extraction algorithm[2] in (Bhati et al., 2017), the modification makes algorithm a little more stable, but only little performance difference is found.

TIMIT (Garofolo et al., 1993) is used for evaluating the phoneme segmentation since it provides ground-truth phoneme boundaries. We follow the setup in (Stan et al., 2016) that uses a small subset of training set as validation to adjust a few algorithm parameters and evaluate on test set. We use a 20ms tolerance window and evaluate with R-value (Räsänen et al., 2009) and precision-recall curve. We hand-pick a visually block diagonal head for each of M3, M6, and M9. We choose MFCC as baseline feature since it is the most prevailing feature (Bhati et al., 2017; 2019; Scharenborg et al., 2010; Mporas et al., 2008) for segmentation. Little performance difference is found between MFCC and $x^0$ (Mel-scale spectrogram).

Fig 2(e) verifies the correlation between block diagonal attentions and phoneme boundaries, that attentions clearly surpass MFCC under the same setting. As for R-value, under the strict hit counting scenario (Räsänen et al., 2009), MFCC achieves 76.68; M3, M6, M9 achieve 79.99, 78.43, 78.19, respectively. Interestingly, larger masking span $C_{num}$ leads to poorer performance. The reason is that when $C_{num} = 3$, masked portions are typically within a phoneme interval, the model learned to utilize features in the same interval to reconstruct. On the other hand, $C_{num} = 9$ can sometimes mask an entire phoneme interval, the model then tries to retrieve information beyond the interval.

Worth mentioning, similarity matrices on MFCCs and learned block diagonal attentions have a fundamental difference that the former show high activation on similar but distant frames in Fig 2(d), while the latter are more aware of phoneme neighborhood structure. Figures similar to Fig 2(d) are shown[2] when we compute similarity matrix on Mel-scale spectrogram or SAT representations, suggesting that despite there are similar frames located far apart, block diagonal heads learned to ignore distant information and focus on neighborhood structure.

## 6. Phoneme Relation Map

To study the functionality of global and vertical heads, we propose to align attentions to phoneme relations to see whether some heads focus on looking for specific phoneme relations in the utterances. For a sequence of input features $x^0$, there exists frame-wise phoneme labels $y \in Y^T$, where $Y$ is a predefined phone set. We consider $x_q^l$ attending to $x_k^l$ as *when observing phoneme $y_q$ the head would look for phoneme $y_k$*. We quantify a phoneme relation $Y_m \rightarrow Y_n$ inside a head $h$ by summing up all attention weights $A_u^h[q, k]$ whose phoneme relation $y_q \rightarrow y_k$ equals $Y_m \rightarrow Y_n$, over the entire speech corpus. More specifically, we plot a phoneme relation map (PRM) $P_h \in \mathbb{R}^{|Y| \times |Y|}$ by the following equations:

$$P_h'[m, n] = \mathop{\mathbb{E}}_{u \sim U} \left[ \frac{1}{T} \sum_{q=1}^{T} \sum_{k=1}^{T} \mathbb{I}_{y_q = Y_m} \cdot \mathbb{I}_{y_k = Y_n} \cdot A_u^h[q, k] \right] \tag{4}$$

$$P_h[m, n] = \frac{P_h'[m, n] - P_U[m, n]}{P_U[m, n]} \tag{5}$$

where $1 \leq m, n \leq |Y|$, $\mathbb{I}$ is indicator function, $P_h', P_U \in \mathbb{R}^{|Y| \times |Y|}$ and $P_U$ is the distribution of all possible phoneme relations[2] in speech corpus $U$, normalizing the effect of dominating relations like $sil \rightarrow sil$ which appears in all utterances. As a result, positive values in $P_h$ represent preference for specific phoneme relations; negative values represent the opposite.

PRMs are plotted using TIMIT (Garofolo et al., 1993) with 39 phonemes, and results of several heads are shown[2] in Fig 3. Since diagonal heads are interpretable themselves, we focus on vertical and global heads. There are several opera-

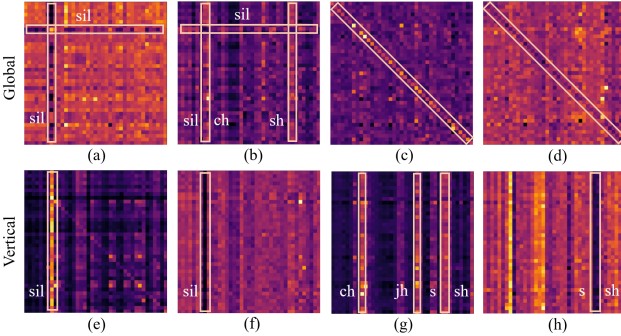

*Figure 3.* Some observed operations from PRMs: (a) sil attends to sil; non-sil attends to non-sil (b) sil attends to non-sil; non-sil attends to sil, ch, sh (c) attends to identity, the same phoneme as query (d) not attends to identity (e) attends to sil (f) not attends to sil (g) attends to ch, jh, s, sh (h) not attends to s, sh.

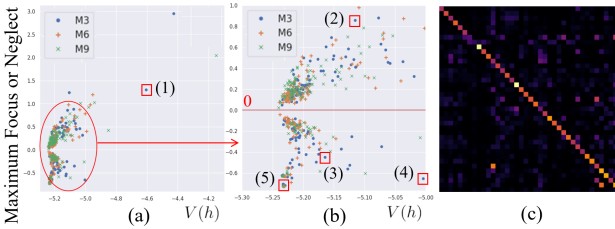

*Figure 4.* (a) The relation between verticality V of a head and its extreme concentration value. Each dot represents a head h. (b) zooms in the bottom-left of (a) since outliers dominate too much. PRMs of representative heads are marked by red squares: Fig 3(e)(g)(f)(h) are for 1,2,3,4 respectively; Fig 4 (c) is for 5.

tions: attending to silence, identity, specific phonemes, and *not* attending to these (*not* operations). We observe tendency of vertical heads either *focus* or *neglect* specific phonemes for all queries, and we bridge their connections. For later discussion, we use *focus* and *neglect* to refer to these types of behaviours. While a PRM characterizes all phoneme relations of a head, we further define concentration $C_h \in \mathbb{R}^{|Y|}$ of a head $h$, where each $C_h[n] \in \mathbb{R}$ quantifies the amount of *focus* (when positive) or *neglect* (when negative) of a head on specific phoneme $Y_n$, over all queries:

$$C_h[n] = \frac{1}{|Y|} \sum_{m=1}^{|Y|} P_h[m, n] \qquad (6)$$

Fig 4 verifies the connection between verticality and concentration. We report the maximum *focus* **or** *neglect* for each head. Fig 4(a) points out that heads with high verticality do *focus* on specific phonemes; Fig 4(b) points out even a slight increase of the verticality V of a head has correlation to concentration, for both *focus* and *neglect*. Some low-verticality heads with extreme *neglect* at the bottom-left of Fig 4(b) are diagonal heads, which always attend to their neighbors dynamically and show extreme *neglect* for all phonemes.

## 7. Importance Ranking

To evaluate the importance of different attention patterns, we conducted two pruning-based probing experiments. We ablate partial functionality of self-attention directly at inference time in two aspects: (1) ablates an entire head; (2) ablates the visible span for all heads. If an attention pattern is essential, ablating it should exhibit immediate loss in terms of the quality of final representations. We examine representation quality by three probing tasks: spectrogram reconstruction, phoneme classification, and speaker recognition. For the first task, we examine the richness in terms of spectrogram details of refined representations. We reuse the reconstruction head during pre-training and measure L1 loss compared to the original. For the latter two tasks, we examine the usefulness of refined representations on downstream tasks. For phoneme and speaker classifications, we train the downstream models using LibriSpeech (Panayotov et al., 2015) train-clean-100 subset and fixed 50k steps. In frame-level setting, we use single-hidden MLP; in utterance-level setting, we use mean-pooling followed by a linear transform. Phoneme classification is conducted under frame-level setting; speaker recognition is under frame-level and utterance-level. Following (Liu et al., 2020), phoneme labels are obtained by the Montreal Force Aligner (McAuliffe et al., 2017), and all evaluations are done on the LibriSpeech test-clean subset.

### 7.1. Head-based Pruning

For each head $h$, we first compute values of $G(h)$, $V(h)$, $D(h)$, and cumulatively prune heads from high to low for each metric by setting $A_u^h = 0$, resulting in three curves as shown in Fig 5(a)(b)(c). We rank the importance of the three categories by observing which pruning results in a larger performance drop. We find ranking results are consistent for different $C_{num}$, so we only show the result of M3. There are several findings: (1) Diagonal heads are the most important. Performances on all three tasks drop significantly with only 24 heads pruned. (2) Vertical heads rank second. While pruning them does not hurt reconstruction or phoneme classification much, it drops faster than global heads in speaker recognition. This suggests that vertical attentions have more relation to speaker identity. (3) Global heads have the least importance that pruning them has the least effect on all tasks. (4) Both global and vertical heads are harmful to the phonetic structure. Fig 5(b) shows that pruning them even improve the phoneme classification accuracy. For vertical heads, we speculate that the vertical heads might *focus* on distant phonemes when forming a new representation independently (disrespectfully) of query phoneme, which might corrupt the local phonetic structure. (5) In Fig 5(b), when we prune according to diagonality, phoneme accuracy drops dramatically for the first 24 heads pruned, while it surprisingly increases as we prune more heads. This

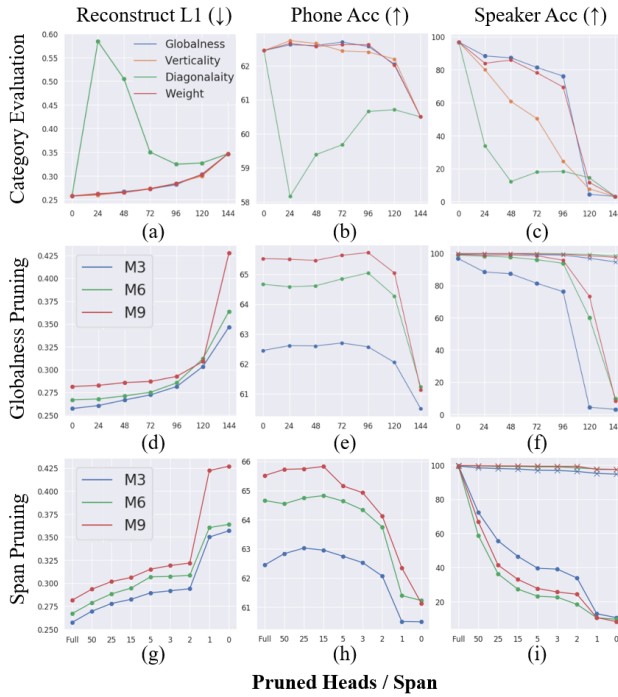

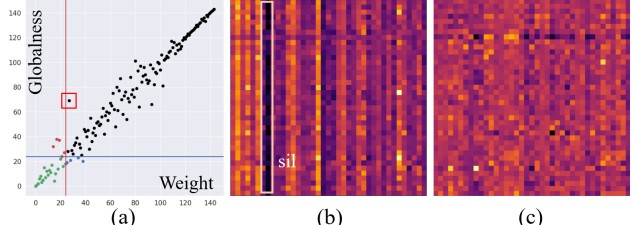

*Figure 6.* (a) Different ranking of a head according to globalness and attention weight. Each dot is a head, and the one with higher ranking number is more important. The first 24 heads to be pruned are green and blue for globalness, green and red for weight. (b) and (c) are PRMs of heads in red and blue dots, respectively.

*Figure 5.* Performance curves of attention pruning. Curves marked by dots are frame-level setting; otherwise utterance-level setting.

is because when pruning more than 24 diagonal heads, we start to prune the heads that are more vertical or global than diagonal, supporting the previous finding that vertical and global attentions are harmful for phonemic information.

We further show the result of ranking the importance of heads based on their maximum attention weights, denoted as *weight* in Fig 5(a)(b)(c), which has been shown to be a strong baseline in the previous work (Hao et al., 2020). Fig 5(c) shows pruning based on globalness has less influence than *weight*. Fig 6(a) visualizes the difference between two ranking strategies. Although they agree on which heads are essential, they slightly diverge on which are not. Their decision boundaries in terms of the first 24 heads to be pruned are shown by red and blue lines in Fig 6(a), which is the direct cause of their performance differences. Globalness prunes blue dots while leaving red dots unpruned; and vise versa for *weight* (while they all prune green dots). Since globalness-based pruning results in a better performance than weight-based pruning, this suggests that heads of red dots are more important than blue dots. We select the head with the highest ranking difference from both red and blue dots, and plot their PRMs in Fig 6(b) and (c), respectively. We find that while Fig 6(b) shows strong *neglect*, (c) does not possess observable operation. In fact, heads of red dots are mostly with clear *neglect*[2]. We argue that this is the main reason why globalness performs better after pruning, that heads with *neglect* are essential to speaker identity, and glob-

alness defined by entropy is able to recognize *neglect* and score them higher. On the other hand, *weight* is confused by attentions with large weights but without meaningful operation, suggesting that *weight* do not always reflect the importance of heads. We speculate that these heads might learn to *neglect* less useful frames, like *sil* in Fig 6(b), and *focus* more on other frames with more speaker information (Wang et al., 2018). Based on the above observations, we choose globalness as our refinement metric. Fig 5(d)(e)(f) show pruning results for M3, M6, M9. The importance of global heads become less for larger $C_{num}$, and we keep observing performance boost for phoneme classification. Despite all three models drop for speaker recognition, the drop is mitigated dramatically in utterance-level setting (a more common scenario), suggesting that global heads are not necessary when speaker classification is performed on utterance level. In conclusion, we can prune SAT heads for more than 50% without sacrificing performance.

### 7.2. Span-based Pruning

Since most of the heads have attention span over a long range (no matter what category it belongs to), we further conduct attention-span pruning to examine if global information is genuinely not helpful for extracting phonetic information. We limit the visible span of all heads by length $r$, either to the left or right. That is, we set $A_u^h[q, k] = 0$ for any $|q - k| > r$. Results are presented in Fig 5(g)(h)(i).

## 8. Conclusion

In this paper, we present multiple strategies for analyzing the self-attention mechanism in SATs, including phoneme segmentation, phoneme relation map, and two aspects of pruning. We find several attention functionality and operations. We identify critical attentions and show our visualization tool useful for understanding pruning behavior. Finally, we conclude that we can refine representations and speed up inference time for a given SAT in two aspects: removing global heads or constraining attention span.

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
