# OpenReview forum: "Understanding Self-Attention of Self-Supervised Audio Transformers"
_ICML.cc/2020/Workshop/SAS — SAS 2020_

### Official Review · AnonReviewer3 · 2020-06-28
**Interesting study and categorisation of self-attention heads in audio transformers**

**Rating:** 8
**Confidence:** 3

**Review:**

This paper provides an interesting study of self-attention heads in audio transformers. They find that self-attention heads tend to converge into three categories (global, vertical, and diagonal), and show extensive evaluations of their respective pros and cons.

The paper is written clearly and provides interesting insights that lead to actionable results. For example, they show that pruning global and vertical heads can improve phoneme classification accuracy. Overall, the paper contributes to a better understanding of self-attention.

My only concern about the paper revolves around its match with this workshop. While the paper claims to work with self-supervised audio transformers, it actually uses a reconstruction error for training, which in my understanding would put it into the field of unsupervised learning. However, since these boundaries are not well defined in general, I did not lower my rating based on this point.

Some (minor) critical points: Some plots miss axis-labels, impairing their understandability. I would recommend putting additional material in an appendix instead of/ in addition to the demo page. A description of the intuitions underlying equations 1-3 would be nice.

---

### Official Review · AnonReviewer2 · 2020-06-29
**Interesting analysis of what self-attention focuses on in self-supervised audio transformers**

**Rating:** 7
**Confidence:** 3

**Review:**

The paper analyses the distribution of the attention weights of multi-attention heads in self-supervised audio trasformers. It first categorizes the weight distributions into 3 main categories: global, vertical and diagonal, and ranks the heads into these categories using entropy-based metrics. After that, the paper analyzes the importance of these head categories with respect to phone and speaker recognition via pruning experiments.
I think that the results presented in the paper are novel, and are important for understanding how multi-head attention works in speech-related tasks. I see no major flaw in the approach of the paper. I found it particularly interesting that the diagonal heads are more important for speech recognition while the vertical heads are more related to speaker recognition - it just coincides with our expectations.
I have only minor remarks that could help improve the quality of the paper:

1. The mathematics in the paper is very dense, the formulas are hard to follow, and they assume strong prior knowledge of the field.  But I don't want to blame the authors for this, as I am not sure if it could be done any better within this short size limit.

2. The figures are terribly small, the figure captions are barely legible. But again, I can accepts this as there is a link to a web page with supplementary material.

3. There are plenty of references, and the are really up-to-date. However, in many cases important details are missing from the references. I don't mind page numbers, but in several cases even the volume (proceedings?) names are missing. Please extend these.

4. Subsection 7.2 is sort of unfinished, At least a couple of sentences would be necessary to help the reader intepret the findings of span-based pruning.

5. All over the paper you talk about phonemes, but I think in most cases phones would be more proper. See Roger Moore's paper "On the Use/Misuse of the Term 'Phoneme'".

6. The English of the paper is quite good, but not perfect. I list just those mistakes where a correction would increase the clarity of the paper:
- "moderately removing global attentions" - this sounds weird to me, as we either remove something or not, "Moderately removing" makes no sense to me. Maybe "partially removing" would sound better.
- "Continously long" occurs two times in the paper, buit I really could not understand what is the "continuously long and locally smooth problem of speech". Please clarify.
- "In the sense of three categories" --> with respect to the three categories
- "Some exhibit highly focused like" --> some exhibit a highly focused behavior like
- "phoneme neighbor structure" --> phonemic neighborhood structure
- "globalness performs better than weight after pruning" --> globalness-based pruning results in a better performance than weight-based pruning
- "we can prune SAT heads for more than 50%" --> we can prune more than 50% of the SAT heads

---

### Decision · Program_Chairs · 2020-07-01

**Decision:**

Accept

**Comment:**

Dear author(s),

Thank you very much for your submission at the ICML2020@SaS workshop (https://icml-sas.gitlab.io/). Based on the scores assigned by the reviewers, we are happy to notify you that your paper was accepted for the workshop.

Please, address the comments of the reviewers and submit the camera-ready version by July 8. We ask the authors to record a 15min video for your talk. At the workshop, we will have the pre-recorded video as well as a live QA session. It is important to keep this time limit, otherwise, your talk will be automatically cut. The deadline for uploading the video is July 8. The detailed instructions for uploading will follow.

Feel free to contact us for any questions!

Best,

The ICML20@SaS organizers:
Mirco Ravanelli
Titouan Parcollet
Dmitriy Serdyuk
Devon Hjelm
Bhuvana Ramabhadran